# Adding Fruit Fermentation Liquid Improves the Efficiency of the Black Soldier Fly in Converting Chicken Manure and Reshapes the Structure of Its Intestinal Microbial Community

**DOI:** 10.3390/insects16050472

**Published:** 2025-04-29

**Authors:** Lifei Chen, Guiying Wang, Hanhan Song, Qi Yang, Jiani Fu, Jiale Liu, Haoyang Sun, Yuxi Wang, Qile Tian, Yuting Sun, Lei Sun, Hao Xin, Zuyin Xiao, Guoliang Wang, Zixuan Zhang, Yinling Zhao, Hongyan Yang, Lusheng Li

**Affiliations:** 1College of Agriculture and Biology, Shandong Province Engineering Research Center of Black Soldier Fly Breeding and Organic Waste Conversion, Liaocheng University, Liaocheng 252000, China; 2Shandong Fengxiang Co., Ltd., Liaocheng 252323, China; 3Liaocheng City Agricultural Technology Extension Service Center, Liaocheng 252000, China

**Keywords:** black soldier fly larvae, fruit fermentation liquid, chicken manure conversion, intestinal microbial community, ammonia emissions

## Abstract

Farmers face challenges in managing chicken manure due to the risk of environmental pollution. This study demonstrates that incorporating fruit fermentation liquid, derived from apples and watermelons, into chicken manure fed to black soldier fly larvae significantly enhanced waste conversion. Larvae receiving the highest dose increased their manure processing efficiency by 9.5%, reduced ammonia emissions by 25%, and promoted the growth of beneficial gut bacteria while suppressing harmful strains. Additionally, the treated manure became safer, less odorous, and more environmentally friendly. This sustainable approach not only enables farmers to recycle waste into high-value insect protein but also supports eco-friendly agricultural practices through reducing reliance on chemical fertilizers.

## 1. Introduction

With the improvement in living standards, people’s demand for meat, eggs, and milk has been increasing, which has led to a continuous increase in the number of livestock and poultry farms [1,2]. As a result, the amount of livestock and poultry manure has also gradually increased. In China, about 3.8 billion tons of livestock and poultry manure are generated each year [3]. Traditional methods for treating manure have been utilized for decades to manage waste, reduce environmental pollution, and recycle nutrients [4]. These conventional practices include composting, anaerobic digestion, direct application to fields, dehydration and drying, incineration, and mechanical separation [5,6,7]. While these methods have been proven to be effective in many cases, each comes with limitations in terms of their efficiency, sustainability, and environmental impact [1,8]. Given the increasing global demand for sustainable farming practices and the need to mitigate the environmental impacts of livestock production, current research is focused on improving these traditional methods or developing novel technologies that integrate waste treatment with resource recovery, such as microbial treatments, insect-based waste management, advanced bioreactor systems, and integrated multi-stage systems [9,10,11].

Insects have gained significant attention in recent years due to their ability to process waste and have become a key aspect in the development of sustainable waste management practices [12]. This capability is largely attributed to their natural enzymatic systems, high reproductive rates, and efficient conversion of organic matter into useful by-products, such as protein, lipids, and chitin, all of which have valuable applications in animal feed, bioplastics, and pharmaceuticals [13,14]. Several insect species, such as *Tenebrio molitor*, *Gryllus,* and *Hermetia illucens*, have emerged as central figures, whereas later species have distinct advantages in terms of their adaptability to various waste streams [10,15,16]. The black soldier fly (BSF) is perhaps the most well-studied insect in the context of organic waste processing. Black soldier fly larvae (BSFL) are capable of converting a wide range of organic waste materials, including food scraps, agricultural by-products, and even certain types of industrial waste [11]. BSFL exhibit high feed conversion efficiency, converting up to 50% of the organic material they consume into body mass. This makes them particularly effective in transforming low-value organic waste into high-value protein-rich biomass, which can be used in animal feed, particularly in the aquaculture and poultry industries [12,17,18,19,20]. Additionally, larvae produce other valuable by-products, including frass (a mixture of feces and exoskeletons), which has been shown to have potential as a natural fertilizer [21]. Moreover, extracts from the exoskeleton of BSFL include promising biopolymers with applications in the biomedical and agricultural sectors, such as wound healing, pest control, and biodegradable plastics [12]. Previous studies have shown that the digestive system of BSFL consists of the foregut, midgut, and hindgut, with the midgut being the primary region responsible for digestion and absorption [17]. The midgut can be further divided into three distinct regions based on pH: the anterior midgut (weakly acidic, pH = 6), the central midgut (strongly acidic, pH = 2), and the posterior midgut (alkaline, pH = 8.5) [22,23,24]. The varying pH values within the midgut lumen play a crucial role in supporting the colonization of gut microbiota, the secretion of digestive enzymes, the degradation of toxic substances, and the dissolution of nutrients, thereby facilitating efficient digestion and nutrient absorption [25].

Research has demonstrated that the addition of micro-organisms significantly influences the efficiency of organic waste conversion by BSFL. Jordan and Tomberlin reviewed the crucial impact of micro-organisms on insect growth, reproduction, and conversion efficiency [26]. Somroo et al. [27] reported that, in experiments where BSFL were co-treated with *Lactobacillus buchneri*, the larval yield, substrate reduction rate, and bio-conversion rate were increased by 37%, 14%, and 38%, respectively, compared to the control group that only used BSFL. Xiao et al. [28] observed that when chicken manure was co-treated with *Bacillus subtilis* and BSFL, the larval weight, bioconversion rate, and substrate reduction rate increased by 16%, 13%, and 13%, respectively. These findings suggest that the addition of micro-organisms results in the production of small-molecule nutrients, such as sugars, peptides, short-chain fatty acids, and non-protein nitrogen, in the substrate matrix, thereby enhancing the growth performance of BSFL and the rate of substrate consumption [29,30]. However, existing research has primarily focused on the effects of adding single microbial strains on the organic waste conversion efficiency of BSFL, with limited studies on the combined use of complex microbial communities and BSFL in organic waste conversion. Fruit fermentation liquid (FFL), which is produced via the anaerobic fermentation of decaying or leftover fruits, contains beneficial probiotics and is rich in organic acids, making it a potential composite microbial solution for organic waste treatment [7,31].

In this study, BSFL were used to process chicken manure in order to explore the effects of different doses of FFL added to chicken manure on the efficiency of BSFL in converting waste; we also explored the emissions of ammonia and hydrogen sulfide, as well as the impact on the gut microbiota of BSFL. This study may provide a potential sustainable and environmentally friendly method to enhance the efficient treatment of organic waste using BSFL.

## 2. Materials and Methods

### 2.1. Materials

The BSF eggs were obtained from Shandong Woneng Agricultural Co., Ltd. (Liaocheng, China). The eggs were incubated at 30 °C and 60% relative humidity for 3 days in an incubator. After hatching, the larvae were reared in a 1000 mL breeding box (bottom diameter 10 cm, top diameter 14.7 cm, height 8.4 cm). During the rearing phase, the humidity was adjusted to 70%, and wheat bran moistened with water was used as feed. The larvae were cultivated in an incubator set at 28 °C with 60% relative humidity. For every 1 g of eggs, approximately 100 g of feed was added, with 50 g added in two separate doses. A total of 50 g of the feed was added on the first day, and the remaining 50 g of the feed was added on the third day. The larvae were reared to the third instar stage, which is 5 to 6 days old, at which point they enter the voracious feeding period. Fresh chicken manure was supplied by Liaocheng Dingshun Breeding Co., Ltd. (Liaocheng, China) and had a moisture content of 80%. The raw materials for the FFL were sourced from a fruit shop located near Liaocheng University. The materials consisted of apples and watermelons, mixed in a weight ratio of 1:1, and supplemented with 0.5% glucose. The mixture underwent anaerobic fermentation for 25 days at 25 °C in a 50 L anaerobic fermentation tank (manufactured by Shanghai Bailun Biological Technology Co., Ltd., Shanghai, China). Following fermentation, the liquid was extracted, filtered through a 100 μm mesh filter, and stored at −20 °C as the standby FFL. The physical and chemical properties of the FFL were determined according to previous research methods [32]. The pH value was 3.51, the total carbon content was 0.04 mg/mL, the total nitrogen content was 5.87 mg/mL, the total acidity content was 14.14 mg/mL, and the total phenols content was 0.31 mg/mL.

### 2.2. Experimental Design and Sample Collection

The experiment was divided into three groups—control group A, experimental group B, and experimental group C—with three replicates per group. Each replicate involved the rearing of BSFL in a 1000 mL plastic box. The feeding substrate treatment consisted of adding different proportions of FFL to fresh chicken manure in a single application. The specific treatment methods were as follows: control group A consisted of 300 g of chicken manure and 50 g of sterile water; experimental group B consisted of 300 g of chicken manure, 25 g of FFL, and 25 g of sterile water; and experimental group C consisted of 300 g of chicken manure and 50 g of FFL. In each breeding box, 300 BSFL were added after the rearing stage. The environmental temperature was set to 28 °C, with a relative humidity of approximately 70%, and the larvae were reared for 7 days.

Every day, the body length and width of 10 randomly selected BSFL in each replicate were measured with a vernier caliper, and their weight changes were recorded with an analytical balance. The weight of the substrate used at the beginning and end of the experiment, as well as the weight of the larvae and their dry weight at the end of the experiment, was recorded for the calculation of the BSFL dry matter conversion rate and the feed-to-larvae ratio. On the 7th day of the experiment, the larvae and chicken manure substrate were separated. Larvae and substrate samples were collected, dried, and ground to obtain BSFL meal samples. These samples were then analyzed for moisture content, crude protein, crude fat, ash content, calcium content, and phosphorus content. Intestinal content samples of BSFL were collected on the 1st, 4th, and 7th days and were labeled as CICd1, CICd4, and CICd7, respectively. As there were no differences in the gut microbiota of *Hermetia illucens* larvae among different groups on the first day, we only collected and sequenced the sample of ICd1A.

Larval Gut Sampling Method: On a UV-sterilized sterile workbench, the gut samples of the BSFL were dissected. The BSFL were removed from the substrate using sterile forceps and then washed with sterile water under high pressure. The larvae were then immersed in 75% ethanol for 2–3 min for surface sterilization. Afterward, the surface of the larvae was washed again with sterile water under high pressure. A sterile dissecting needle was used to enter the gut of the BSFL, which was extracted and then placed in a sterile 5 mL screw-cap cryotube. Each sample was frozen in liquid nitrogen for 15 min and stored at −80 °C for further analysis. Chicken manure substrate samples were collected on the 1st, 4th, and 7th days and were marked as CMd1, CMd4, and CMd7, respectively.

Substrate Sampling Method: Using a sterile high-pressure scoop, the substrate was extracted and placed into a sterile 5 mL screw-cap cryotube. The sample was frozen in liquid nitrogen for 15 min and stored at −80 °C.

### 2.3. Analysis Methods

#### 2.3.1. Measurement of BSFL Growth Performance and Nutritional Composition

Body Length: The body length of randomly selected larvae (10 larvae per replicate, 30 larvae per group) was measured daily using a Vernier caliper.

Body Width: Body width (typically measured at the widest point of each larva’s body) was measured daily.

Larval Weight: The weight of the larvae was recorded daily using an analytical balance. This allows for tracking of larval growth over the rearing period.

Dry Weight: After the 7-day rearing period, the larvae were dried at 105 °C until a constant weight was achieved, and the dry weight was recorded.

Dry Matter Conversion Rate (DMCR): The dry matter conversion rate was calculated as the ratio of the dry weight of the larvae at the end of the experiment to the dry weight of the substrate (chicken manure + FFL) used.$$DMCR(\%)=\frac{Dry\;Weight\;of\;Larvae}{Dry\;Weight\;of\;Substrate}\times 100$$

Feed conversion rate (F/C) refers to the ratio between the total weight of chicken manure and the total weight of the BSFL at the end of the experiment. It reflects the biomass produced by the BSFL during the conversion of chicken manure and serves as a measure of the efficiency of the process. This ratio is typically used to assess the ability of BSFL to convert organic waste (such as chicken manure) into biomass and to evaluate their growth performance.$$F/C(\%)=\frac{Total\;weight\;of\;chicken\;manure}{Total\;weight\;of\;BSFL\;at\;the\;end\;of\;the\;experiment}\times 100$$

The determination of moisture, crude protein, crude fat, crude ash, calcium content, and phosphorus content in BSFL was conducted in accordance with the Chinese National Standards GB/T 6435-2014 [32], GB/T 6432-2018 [33], GB/T 6433-2006 [34], GB/T 6434-2006 [35], GB/T 6436-2018 [36], and GB/T 6437-2018 [37], respectively.

#### 2.3.2. Determination of Ammonia (NH_3_) and Hydrogen Sulfide (H_2_S)

Ammonia emissions during the conversion process with BSFL in each rearing box were measured daily using the Chinese National Standard GB/T 14679-1993 [38]. On the 4th and 7th days of the experiment, hydrogen sulfide emissions during the conversion process with BSFL were measured according to the Chinese National Standard GB/T 14678-1993 [39].

#### 2.3.3. DNA Extraction, PCR Amplification, and Sequencing

Samples of the substrate and BSFL gut microbiota were collected on day 1, day 4, and day 7. Total DNA was extracted from these samples using the PowerSoil^®^ DNA Isolation Kit (MO BIO Laboratories, San Diego, CA, USA). The bacterial 16S rRNA gene V3–V4 hypervariable region was amplified by PCR using the primer pair 338F-806R, with sequences 338F (5′-ACTCCTACGGGAGGCAGCAG-3′) and 806R (5′-GGACTACHVGGGTWTCTAAT-3′). The PCR amplification conditions were as follows: initial denaturation at 95 °C for 3 min; 27 cycles of denaturation at 95 °C for 30 s, annealing at 55 °C for 30 s, and extension at 72 °C for 45 s; followed by final extension at 72 °C for 10 min. PCR products were analyzed by using 2% agarose gel electrophoresis. The gel was extracted using the Omega Gel Extraction Kit (Omega Bio-tek, Norcross, GA, USA), and the PCR products were recovered by elution with Tris-HCl. Recovery was further confirmed by electrophoresis on 2% agarose gel. The PCR products were quantified using a NanoDrop 2000 spectrophotometer (ThermoFisher Scientific, Waltham, MA, USA). After determining the concentration of each sample, the PCR products were mixed in appropriate proportions according to sequencing requirements and sent to Shanghai Yuanxun Biotechnology Co., Ltd. (Shanghai, China) for sequencing using the Illumina Miseq platform (Illumina Inc., San Diego, CA, USA). High-fidelity (HiFi) reads were obtained from the subreads, generated using circular consensus sequencing via SMRT Link v11.0.

#### 2.3.4. Bioinformatics Analysis

The raw sequencing data were processed through a series of bioinformatics analyses. First, denoising was performed using the exact sequence variant method [40]. The process of generating Amplicon Sequence Variants (ASVs) begins with merging paired-end reads using tools like VSEARCH v2.21.1 (with the following parameters: min overlap = 20, max mismatch = 0.2), followed by rigorous quality filtering and primer removal via Cutadapt v4.4 (Q20/Q30 thresholds, excluding reads with >5% ambiguous bases). Chimeric sequences are then eliminated using VSEARCH v2.21.1 in reference-based (--uchime_ref), leveraging databases such as SILVA v138 [41]. Denoising is performed using DADA2 v1.28 (error correction and chimera removal) and Deblur v1.1.0 (greedy error-trimming for single-end reads). Taxonomic annotation of ASVs relies on SILVA v138 databases, classified through QIIME2 v2023.2’s feature classifier, culminating in an ASV abundance table formatted as BIOM v2.1 or CSV for downstream ecological or statistical analyses [42]. The ASV table was manually filtered, i.e., chloroplast sequences in all samples were removed. To minimize the effects of sequencing depth on alpha and beta diversity measures, the number of 16S rRNA gene sequences from each sample was rarefied to 6000, which still yielded an average Good’s coverage of 99.09%. To assess microbial community diversity, α-diversity indices, including community richness (Chao1) and community diversity (Simpson and Shannon), were calculated using QIIME 2 and visualized with R version 4.4.3 (available at http://cran.r-project.org/src/base/R-4/, accessed on 12 October 2024). The relative abundance of microbial taxa was displayed as stacked columns. For β-diversity analysis, principal coordinate analysis (PcoA) based on weighted UniFrac distance was employed to visualize differences among OTU profiles. Taxonomic cladograms were constructed to further illustrate the phylogenetic relationships among OTUs. To identify significantly different bacterial taxa (biomarkers) among groups, linear discriminant analysis effect size (LefSe) was conducted based on the Kruskal–Wallis qsum-rank test [43]. Additionally, linear discriminant analysis (LDA) was used to estimate the effect size of each biomarker (LDA score > 2, *p* < 0.05). Spearman’s correlation coefficients between probiotics and ammonifiers were calculated using the psych package in R version 4.4.3 [44]. These correlations were calculated at the level of means, and the Holm–Bonferroni method was used for correction. A correlation between two nodes was considered statistically robust if the Spearman’s correlation coefficient was greater than 0.6 or less than −0.6 and if the *p*-value was less than 0.01.

### 2.4. Statistical Analysis of Data

The experimental data were processed using Microsoft Excel 2016 software (Microsoft Corporation, Redmond, Washington, DC, USA) and then subjected to significance testing using SPSS 26.0 (IBM, Armonk, NY, USA). The results were expressed as the mean ± standard error (n = 3). A Linear Mixed Model (LMM) was employed to account for within-subject correlations. Specifically, the “individual larvae/replicate boxes” were incorporated as a random effect (e.g., larvae were randomly selected from replicate boxes), thereby allowing repeated measurements on the same subjects (larvae or boxes) across time points (Days 1, 4, and 7) to be controlled. Student’s *t*-test and two-way ANOVA with post hoc Tukey tests were used to test whether there were differences among treatments after verifying normality (Shapiro test) and homogeneity of variances, with *p* < 0.01 considered highly significant and *p* < 0.05 considered significant, both annotated with uppercase and lowercase letters and asterisks. Bar graphs were generated using GraphPad Prism 8 software (GraphPad Software, Inc., San Diego, CA, USA).

## 3. Results

### 3.1. The Effect of the Addition of FFL on the Growth Performance of BSFL

As shown in Figure 1a–c, there were no significant differences in the body length, body width, and body weight of BSFL between the experimental groups B and C and the control group A at any age (*p* > 0.05). In terms of the body length growth trend, the larvae in all groups exhibited the most rapid increase in body length on day 3. From day 6 onwards, the body length of the larvae ceased to increase and slightly decreased. In terms of the body width variation trend, the body width of BSFL between the experimental groups B and C and the control group A at any age showed an initial increase followed by a decrease. Starting from day 5 of the experiment, the body width of the larvae in the control group began to decrease. From day 6 onwards, the body width of larvae in experimental groups B and C also began to decrease. In terms of the body weight variation trend, the body weight of BSFL in all groups generally increased over time. These results indicate that the addition of FFL did not significantly affect the body length, body width, and body weight variation in BSFL during the conversion of chicken manure. As shown in Figure 1d,e, compared to control group A, the dry matter conversion rate in test group C increased by 9.5%, with a significant difference (*p* < 0.05), and the feed-to-insect ratio decreased significantly by 1.02 (*p* < 0.01). Test group B showed a trend of improvement in both dry matter conversion rate and feed-to-insect ratio compared to control group A, but the differences were not significant (*p* > 0.05). No significant differences were observed between test group B and test group C for both indicators (*p* > 0.05). These results indicate that the addition of FFL can improve the conversion efficiency of chicken manure by black soldier flies. In terms of dosage, the effect was more pronounced with the addition of 50 g of FFL.

### 3.2. The Effect of Adding FFL on the Release of NH_3_ and H_2_S

As shown in Figure 2a, the NH_3_ emissions from control group A, experimental group B, and experimental group C all exhibited an initial increase followed by a decrease. On day 5 of the experiment, the NH_3_ emissions reached their peak. Compared to the control group, the NH_3_ emissions from experimental group C and experimental group B decreased by 24.48 and 20.67 mg kg^−1^ DM day^−1^, respectively, with significant differences (*p* < 0.01). The NH_3_ emissions from experimental group C were 3.81 mg kg^−1^ DM day^−1^ lower than those from experimental group B, with a significant difference (*p* < 0.05). These results indicate that the addition of FFL reduces NH_3_ emissions during the BSFL conversion of chicken manure. In terms of dosage, the addition of 50 g of FFL showed more pronounced effects. As shown in Figure 2b, on day 4 of the experiment, the H_2_S emissions from experimental groups B and C were 0.33 and 1.21 mg g^−1^ DM day^−1^ lower than those from control group A, respectively, with no significant difference (*p* > 0.05). On day 7, the H_2_S emissions from experimental group B were 0.07 mg g^−1^ DM day^−1^ higher than those from the control group, with no significant difference (*p* > 0.05). Additionally, the H_2_S emissions from experimental group C were 0.88 and 0.81 mg g^−1^ DM day^−1^ lower than those from experimental group B and the control group, respectively, with no significant difference (*p* > 0.05). These results suggest that the addition of FFL had no significant effect on H_2_S emissions during the BSFL conversion of chicken manure.

### 3.3. The Effect of Adding FFL on the Gut Microbiota of BSFL

A total of 21 samples from BSFL guts were sequenced and optimized. The number of effective sequences per sample ranged from 30,151 to 118,231. At a 97% similarity threshold, operational taxonomic units (OTUs) were assigned, resulting in the annotation of 4 bacterial phyla, 5 classes, 15 orders, 30 families, 50 genera, and 65 species in the BSFL gut. The samples were considered reasonable for subsequent α-diversity analysis. Chao1 and ACE represent species richness, while Shannon and Simpson indices reflect species diversity. As shown in Figure 3, the species richness and diversity of the gut microbiota in groups A, B, and C of BSFL exhibited a trend of initially increasing and then decreasing, with the highest species richness and diversity observed on day 4 of the experiment. On both day 4 and day 7, the species richness and diversity of the gut microbiota in groups B and C were higher than in control group A. These results suggest that the species richness and diversity of the BSFL gut microbiota follow a trend of increasing initially and then decreasing, and the addition of FFL can enhance the species diversity and richness of the gut microbiota. The addition of 50 g of FFL had a more significant impact on microbial species diversity.

As shown in Figure 4a, the intestinal microbiota of BSFL across three experimental groups was dominated by four phyla: Proteobacteria, Firmicutes, Bacteroidota, and Actinobacteriota. Significant temporal and treatment-dependent shifts were observed (LefSe LDA >2.0, FDR-adjusted *p* <0.05): on day 4, compared to the control (group A), Proteobacteria decreased by 4.98% (*p* = 0.013) in group B and 6.07% (*p* = 0.007) in group C, while Actinobacteriota increased by 2.19% (*p* = 0.041) and 4.68% (*p* = 0.002), respectively; concurrently, group C exhibited a 4.75% increase in Firmicutes (*p* = 0.038) and a 3.43% decrease in Bacteroidota (*p* = 0.029). By day 7, Proteobacteria rebounded significantly in groups B (+17.21%, *p* = 0.001) and C (+14.52%, *p* = 0.003), whereas Bacteroidota declined sharply (group B: −11.82%, *p* = 0.004; group C: −16.60%, *p* < 0.001), with Firmicutes in group C showing a modest 2.36% increase (*p* = 0.048). At the genus level (Kruskal–Wallis with Benjamini–Hochberg correction), notable changes included: on day 4, group C displayed a reduced abundance of Paenalcaligenes (−7.46%, *p* = 0.011) and Sphingomonas (−7.27%, *p* = 0.009) alongside increases in Corynebacterium (+4.76%, *p* = 0.032) and Gallicola (+4.03%, *p* = 0.025), while group B showed declines in Paenalcaligenes (−3.56%, *p* = 0.047), Oceanisphaera (−6.87%, *p* = 0.002), and Acinetobacter (−4.94%, *p* = 0.015) with a concurrent rise in Gallicola (+3.32%,*p* = 0.042); by day 7, group C demonstrated a marked decrease in Sphingobacteriaceae (−21.16%, *p* < 0.001) but increases in Paenalcaligenes (+3.71%, *p* = 0.038), Providencia (+7.07%, *p* = 0.006), and Ignatzschineria (+4.16%, *p* = 0.018), whereas group B exhibited reductions in Paenalcaligenes (−3.39%, *p* = 0.049), Providencia (−6.10%, *p* = 0.012), and Sphingomonas (−13.76%, *p* < 0.001) alongside a striking 27.63% surge in Morganella (*p* < 0.001).

### 3.4. The Effect of Adding FFL on the Substrate Microbial Community

The alpha diversity of the substrate microbiota during BSFL conversion of chicken manure is shown in Figure 5. On day 1 of the experiment, the microbial species richness and diversity of the substrate in treatment group C were higher than those in control group A and treatment group B. On day 4, the microbial species richness of control group A and treatment group C was higher than that of treatment group B. On day 7, treatment group C exhibited higher microbial species richness and diversity compared to treatment group B, which in turn were higher than those of control group A. These results suggest that the addition of 50 g of FFL enhances the microbial species richness and diversity in the substrate during BSFL-mediated conversion of chicken manure.

As shown in Figure 6a, the substrate microbiota across groups was dominated by Proteobacteria, Firmicutes, Bacteroidota, and Actinobacteriota. While minimal differences were observed on day 1, significant treatment effects emerged by day 4 (LefSe LDA >2, FDR-adjusted *p* < 0.05): in groups B and C versus control A, Proteobacteria increased by 11.13% (*p* = 0.008) and 17.13% (*p* < 0.001), respectively, with concurrent Firmicutes rises of 7.21% (*p* = 0.022) and 3.61% (*p* = 0.048), whereas Bacteroidota decreased markedly (−17.48%, *p* = 0.003; −20.05%, *p* < 0.001). These trends persisted on day 7, with Proteobacteria remaining elevated (group B: +3.82%, *p* = 0.041; group C: +11.07%, *p* = 0.001) and Bacteroidota further declining (−8.36%, *p* < 0.001 in group C). At the genus level (Kruskal–Wallis with Benjamini–Hochberg correction), group C exhibited pronounced reductions in Ulvibacter (−8.75%, *p* = 0.007), Sphingomonas (−6.75%, *p* = 0.012), and Pseudomonas (−5.84%, *p* = 0.009) by day 4, alongside increases in Acinetobacter (+2.33%, *p* = 0.042) and Ignatzschineria (+5.70%, *p* < 0.001); by day 7, Sphingomonas (−11.66%, *p* < 0.001) and Bacillus (−9.04%, *p* = 0.005) continued to decline, while Acinetobacter increased (+2.32%, *p* = 0.047). Group B showed similar but attenuated shifts (e.g., day 4: Ulvibacter −6.07%, *p* = 0.025; Erysipelothrix +4.31%, *p* = 0.018).

### 3.5. The Correlation Analysis of Environmental Parameters and the Abundance of Dominant Microbiota

As shown in Figure 7a, within the intestinal microbiota, the relative abundance of Morganella, Enterococcus, Providencia, and Ulvibacter shows a significant positive correlation with the crude fat content and feed conversion ratio of BSFL, while exhibiting a negative correlation with NH_3_ emissions. On the other hand, the relative abundance of Acinetobacter, Paenalcaligenes, Oceanisphaera, Ignatzschineria, and Gallicola is significantly negatively correlated with BSFL crude fat content and feed conversion ratio but positively correlated with NH_3_ emissions. In the substrate microbiota (Figure 7b), the relative abundance of Ulvibacter, Thiopseudomonas, and Amphibacillus is significantly positively correlated with BSFL crude fat content, feed conversion ratio, NH_3_ emissions, and H_2_S emissions. In contrast, the relative abundance of Erysipelothrix, Paenalcaligenes, Ignatzschineria, Sporosarcina, Oceanisphaera, and Acinetobacter is negatively correlated with BSFL H_2_S emissions, crude fat content, and feed conversion ratio. Furthermore, the relative abundance of Sporosarcina, Oceanisphaera, and Acinetobacter is inversely correlated with NH_3_ emissions.

## 4. Discussion

The conversion of livestock and poultry manure, kitchen waste, and other organic waste by black soldier flies has been adopted as a novel, efficient, and green waste conversion method [45,46]. Previous research has mainly focused on the effects of temperature, moisture content, carbon-to-nitrogen ratio, and different substrate compositions on BSFL’s ability to convert organic waste [21,47]. However, studies exploring the influence of micro-organisms, especially FFL, on the larvae’s conversion of chicken manure are relatively limited. In this study, different doses of FFL were added to investigate the effects on the growth performance of BSFL, the feed conversion rate, NH_3_ and H_2_S emissions, and the microbial communities in the larvae’s gut and the chicken manure substrate during the conversion process of chicken manure. This study shows that the addition of 50 g of FFL significantly improved the dry matter conversion rate of BSFL for chicken manure, while the F/C ratio was significantly reduced. The results show that compared with the control group A, the feed-to-insect ratio of experimental group C decreased, while the dry matter conversion rate increased, and the weight of the insects also increased. This indicates that the consumption of the larvae did not decrease, and thus the resource utilization rate of the manure was higher. This suggests that the addition of FFL facilitates the feeding and digestion of chicken manure by BSFL. It was speculated that this may be due to the low pH of the FFL, which is conducive to the growth and metabolism of probiotics such as *Ignatzschineria*, *Sphingobacteriaceae*, and *Acinetobacter*, thereby enriching the FFL with a large amount of enzymes, vitamins, amino acids, minerals, and other secondary metabolites [31,48,49]. Marcel et al. [50] reported that feeding BSFL with mixed fruit and vegetable waste not only provides macronutrients but also offers abundant micronutrients, thereby improving the growth performance and nutritional content of the larvae. Jucker et al. [51] reported that BSFL fed with fruits had an increased fat content, especially saturated fatty acids. Moreover, the addition of FFL or micro-organisms altered the gut microbiota structure of BSFL, enhancing BSFL gut’s ability to digest and absorb nutrients [26]. Somroo et al. [27] reported that the addition of *Lactobacillus buchneri* L3–9 significantly increased the dry matter conversion rate of tofu dregs by BSFL, as well as the crude protein and crude fat content in the larvae. However, the feed conversion rate significantly decreased, which is consistent with the results of this study. Therefore, the addition of FFL may change the intestinal microbial community structure of BSFL, resulting in a significant increase in body weight [52,53,54].

Emissions of ammonia and hydrogen sulfide are the main sources of odorous pollution in the process of livestock and poultry manure treatment, and they affect people’s normal life and health [55,56,57]. Therefore, reducing odorous emissions during the process of BSFL conversion of livestock and poultry manure is a problem that researchers have been continually trying to solve. Studies have shown that during the process of BSFL conversion of feces, a high pH value will lead to the emission of ammonia gas, while a relatively low pH is not conducive to the release of ammonia gas [58,59,60]. Group C exhibited a 24.48% reduction in NH_3_ emissions compared to group A (*p* < 0.01), while H_2_S emissions remained unchanged. This differential response likely reflects the specific inhibitory effects of fermentation-derived organic acids on ammonia-producing bacteria. The pH of the substrate in group C decreased by 1.2 units (from 6.8 to 5.6), creating an acidic environment that suppresses urease activity and ammonia volatilization [61]. Similarly, the addition of organic acids has been shown to reduce NH_3_ emissions by 57% in swine manure composting [62]. The absence of significant H_2_S emissions reductions suggests that sulfur-reducing bacteria (SRB) in the substrate were not substantially inhibited by the fermentation liquid. This may be due to the limited availability of sulfate ions in chicken manure [63] or the tolerance of SRB to low-pH conditions [64]. Notably, H_2_S emissions in group C were lower on days 4 and 7 compared to groups A and B, indicating a delayed but sustained inhibitory effect. Further analysis of the sulfate metabolism pathway in the substrate microbiome could clarify this phenomenon.

The gut microbiome of group C exhibited significant shifts in bacterial diversity and composition. On day 4, the abundance of Proteobacteria decreased by 6.07%, while Actinobacteriota increased by 4.68%. These changes suggest that FFL promoted the growth of beneficial gut microbes involved in protein degradation and vitamin synthesis (e.g., *Bacillus* and *Streptomyces*) [65]. Conversely, the increase in Proteobacteria on day 7 may reflect adaptive metabolic adjustments to the altered substrate composition. The genus-level analysis revealed a 7.46% decrease in *Paenalcaligenes* (a potential pathogen) and a 4.03% increase in *Gallicola* (a cellulolytic bacterium) in group C on day 4. These findings imply that the FFL selectively enriched beneficial gut taxa that enhance nutrient extraction and suppress harmful micro-organisms [66]. Similarly, the substrate microbiome in group C showed a 17.13% increase in Proteobacteria on day 4, which may contribute to nitrogen mineralization and odor control [67]. The correlations between microbial taxa and NH_3_/H_2_S emissions highlight the critical role of *Acinetobacter* and *Ulvibacter* in modulating odor production. These findings are consistent with previous studies linking *Acinetobacter* to the degradation of organic acids and *Ulvibacter* to the reduction of sulfur compounds [24].

The dual functions of FFL—enhancing larval nutrition and inhibiting odor-producing bacteria—appear to be mediated by its low pH and organic acid content. The fermentation process generates lactic, acetic, and citric acids, which act as antimicrobial agents against ammonia-producing (APB) and SRB [63,68]. Additionally, the pre-digestion of organic matter by FFL may simplify the larval digestive process, allowing for more efficient nutrient absorption [69]. The restructuring of the gut microbiome likely involves horizontal gene transfer and metabolic cross-talk between larvae and microbes. The enrichment of *Ignatzschineria* and *Sphingobacteriaceae* in group C suggests that these taxa play key roles in lipid metabolism and detoxification [70]. Furthermore, the increased Shannon diversity in the gut microbiome indicates a more stable and resilient microbial community, which is critical for larval health and performance under fluctuating environmental conditions [29].

This study demonstrates that low-cost FFL can simultaneously improve the efficiency of BSFL composting and mitigate odor pollution from chicken manure. The 9.5% increase in the DMCR translates to higher biogas yields and reduced waste volume, while the 24.48% NH_3_ emissions reduction aligns with air quality standards for livestock farming areas [71]. In the future, as in previous studies, we can make use of agricultural waste (e.g., apple and watermelon peels) as feedstock for fermentation, promoting circular economy principles [51]. However, large-scale implementation will require optimization of fermentation parameters (e.g., carbon/nitrogen ratio) and larval feeding strategies. Future studies should also investigate the long-term stability of microbial communities and the potential for pathogen transmission in treated substrates.

While this study provides valuable insights into the effects of FFL on *H. illucens*-mediated manure management, several limitations remain. First, the short experimental duration (7 days) may not capture the full developmental and microbial succession patterns of the larvae. Second, the specific functional genes (e.g., urease and hydrogen sulfide synthase) responsible for odor reduction were not directly analyzed. Third, the economic feasibility of scaling up the fermentation process needs to be evaluated. Fourth, it should be noted that baseline microbiota profiling was performed only for the control group (group A) prior to FFL treatment. Although the larvae in all groups originated from the same population and were maintained under uniform conditions, the absence of day 1 sequencing data for groups B and C leaves a possibility of undetected initial microbial variations. Future research should focus on the following: (1) isolating and characterizing key microbial strains from the fermentation liquid responsible for odor suppression and nutrient enhancement; (2) conducting life-cycle assessments to compare the environmental impacts of this method with conventional manure treatment technologies; (3) exploring the synergistic effects of combining FFL with other additives (e.g., biochar or enzymes); and (4) incorporating comprehensive baseline sequencing across all experimental groups to confirm the causality of FFL-induced microbiota shifts. The above four aspects will be addressed in our next study.

## 5. Conclusions

This study demonstrated that adding FFL significantly enhances the conversion efficiency of chicken manure by BSFL and optimizes the microbial community structure in both their guts and the substrate. The high-dose FFL group (50 g) exhibited a dry matter conversion rate, which was increased by 9.5%, a feed-to-larvae ratio, which was reduced by 1.02, and peak ammonia emissions that were decreased by 24.48 mg·kg^−1^·DM·day^−1^, with suppressed hydrogen sulfide release. Microbial analysis revealed that FFL reshaped the larval gut environment through reducing the abundance of Proteobacteria (6.07% decrease) while enriching Actinobacteriota and beneficial genera (e.g., *Corynebacterium*, *Gallicola)*, thus enhancing nutrient absorption. In the substrate, potential pathogens (e.g., *Sphingobacteriaceae*) decreased by 21.16%, while functional taxa (Proteobacteria, Firmicutes) proliferated, indicating that FFL metabolites (e.g., organic acids, small-molecule sugars) inhibited harmful microbes and accelerated organic decomposition. This approach improves the efficiency of waste valorization, reduces greenhouse gas emissions, and offers an innovative, low-cost, and eco-friendly solution for agricultural waste management. Future research should focus on optimizing FFL formulations and scaling up applications to advance circular agriculture systems and achieve carbon neutrality goals.

## Figures and Tables

**Figure 1 insects-16-00472-f001:**
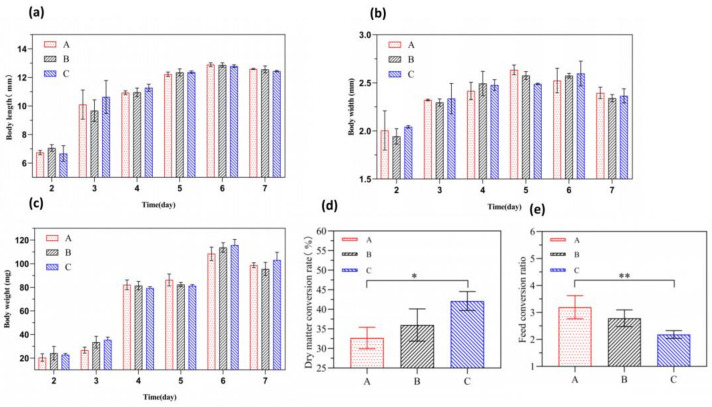
The effect of the addition of FFL on the growth performance of BSFL. (**a**) Body length; (**b**) Body width; (**c**) Body weight; (**d**) Dry matter conversion rate; (**e**) Feed conversion ratio. The results are expressed as the mean ± standard error (n = 3). * *p* < 0.05; ** *p* < 0.01.

**Figure 2 insects-16-00472-f002:**
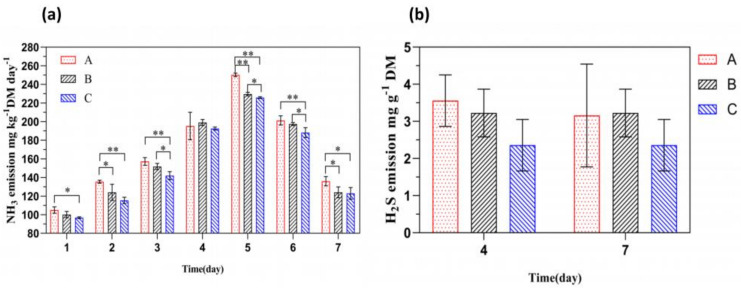
The effect of adding FFL on the release of NH_3_ and H_2_S. (**a**) NH_3_ emission; (**b**) H_2_S emission. The results are expressed as the mean ± standard error (n = 3). * *p* < 0.05; ** *p* < 0.01.

**Figure 3 insects-16-00472-f003:**
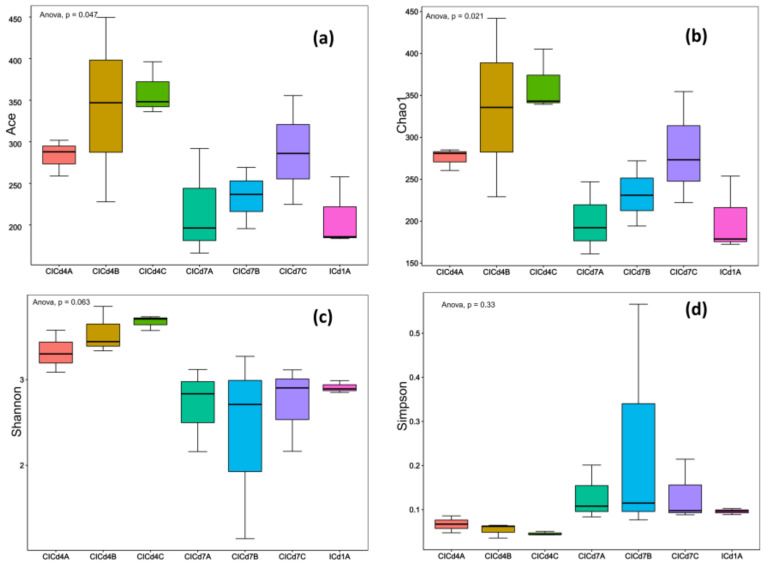
Analysis of the alpha diversity of intestinal micro-organisms in BSFL after adding FFL. (**a**) Chao1 index, (**b**) Ace index, (**c**) Shannon index, (**d**) and Simpson index.

**Figure 4 insects-16-00472-f004:**
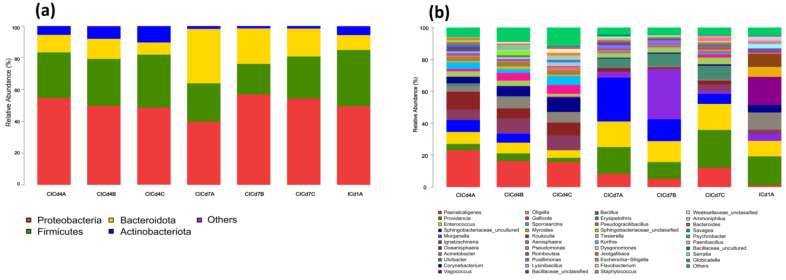
Analysis of the intestinal microbial community structure of BSFL after adding FFL: (**a**) at the phylum level and (**b**) at the genus level.

**Figure 5 insects-16-00472-f005:**
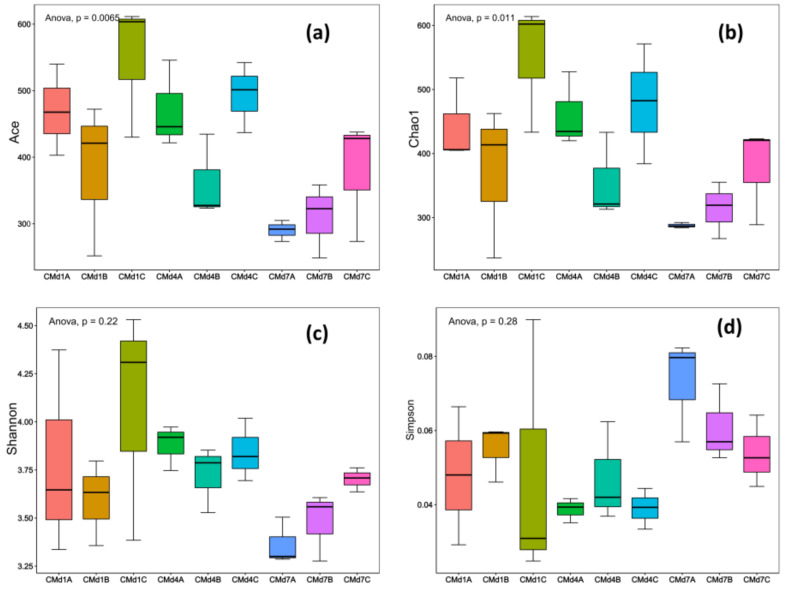
Analysis of microbial alpha diversity in chicken manure substrate after adding FFL. (**a**) Chao1 index, (**b**) Ace index, (**c**) Shannon index, and (**d**) Simpson index.

**Figure 6 insects-16-00472-f006:**
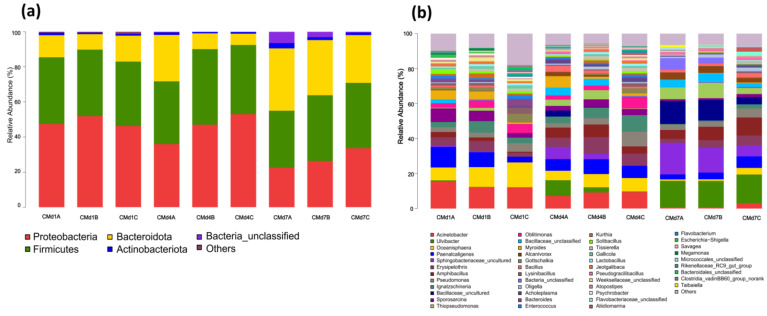
Analysis of the microbial community structure of chicken manure substrate after adding FFL: (**a**) at phylum level and (**b**) at the genus level.

**Figure 7 insects-16-00472-f007:**
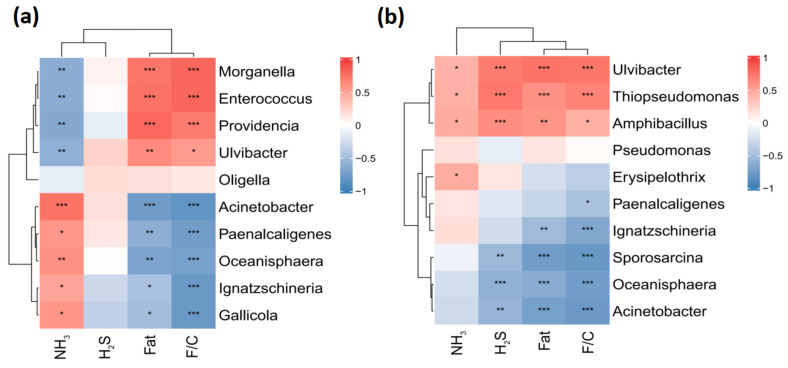
Correlation analysis of environmental parameters and the abundance of dominant microbiota based on Spearman’s heat map. (**a**) Spearman’s correlation coefficient heat map of different influencing factors and the top 10 dominant microbiota in the gut microbiota of BSFL; (**b**) Spearman’s correlation coefficient heat map of different environmental factors and the top 10 dominant microbiota in the gut microbiota of chicken manure substrate. * *p* < 0.05, ** *p* < 0.01, and *** *p* < 0.01.

## Data Availability

The data presented in this study are available upon request from the corresponding author. The data are not publicly available due to company policies.

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
