# Peer review of "Adding Fruit Fermentation Liquid Improves the Efficiency of the Black Soldier Fly in Converting Chicken Manure and Reshapes the Structure of Its Intestinal Microbial Community"

_insects, 2025, doi:10.3390/insects16050472_

Round 1
Reviewer 1 Report
Comments and Suggestions for Authors
Dear Authors,
The work you've done is interesting. In the attached file, I've highlighted some things I think should be corrected. In addition, I think that the results of the study are valuable considering the production of BSF but not the reduction in manure, which is the problem originally raised by the study. You should reconsider this situation. I also think it's important to review the wording of the materials and methods section because it's not clear whether the work was done using the fresh or dry weight of the manure.

The authors should review the verb tenses in the text because it mixes present, future, and past. The text must be written in the past tense.
Author Response
Comments 1: [If the document does not address the use of the byproduct as fertilizer, it should be raised as a possibility.] |
Response 1: Thank you for pointing this out. We have made corresponding revisions according to your comments and those of other reviewers. Revised version line 13-22: Simple summary section has been changed to “Farmers face challenges in managing chicken manure, which poses a risk of environmental pollution. This study demonstrated that incorporating fruit fermentation liquid, derived from apples and watermelons, into chicken manure fed to black soldier fly larvae significantly enhanced waste conversion. Larvae receiving the highest dose increased manure processing efficiency by 9.5%, reduced ammonia emissions by 25%, and promoted the growth of beneficial gut bacteria while suppressing harmful strains. Additionally, the treated manure became safer, less odorous, and environmentally friendlier. This sustainable approach not only enables farmers to recycle waste into high-value insect protein but also supports eco-friendly agricultural practices by reducing reliance on chemical fertilizers.”. |
Comments 2: [delete resource recovery] |
Response 2: Agree. Thank you for pointing this out. We have made corresponding revisions according to your comments and those of other reviewers. Revised line 57-59: The sentece has been changed to “integrate waste treatment with resource recovery like: microbial treatments, insect-based waste management, advanced bioreactor systems, and integrated multi-stage systems [9-11]. ”.
Comments 3: [efficiency, sustainability] Response 3: Thank you for your comments. We have made corresponding revisions according to your comments and those of other reviewers. We have deleted the constent in revised line 59.
Comments 4: [L111: were obtained.] Response 4: We have made corresponding revisions according to your comments
Comments 5: [L112: delete in an incubator] Response 5: Thank you for your comments. We have made corresponding revisions according to your comments and those of other reviewers. The sentence has been changed to “The eggs were incubatedat 30°C and 60% relative humidity for 3 days in an incubator.”.
Comments 6: [L113: after hatching?] Response 6: Thank you for pointing this out. We have changed to “After hatching”.
Comments 7: [L115,L117: delete artificial] Response 7: Thank you for your comments. We have deleted it.
Comments 8: [L117-118:why did you use tho doses? when did you add the fist and the second? ] Response 8: Thank you for your comments. They are added separately to maintain air permeability. 50 grams of the feed is added on the first day, and the remaining 50 grams of the feed is added on the third day. The sentence “50 g of the feed is added on the first day, and the remaining 50 g of the feed is added on the third day.” was added.
Comments 9: [L121-122: I think it is important that you indicate the variety and ripeness of the fruit used. It's also important that you explain why glucose is added.] Response 9: Thank you for your comments. Adding glucose can quickly initiate the fermentation process and enable microorganisms to grow rapidly.
Comments 10: [L126: the congelation is an deodorant method? this is not very clear in the text.] Response 10: Thank you for your comments. “deodorant” has been changed to “standby”.
Comments 11: [L134: delete which is fed into the plastic box] Response 11: Thank you for your comments. The sentence has been changed to “The feeding substrate treatment consists of adding different proportions of FFL to fresh chicken manure in a single application.”
Comments 12: [L141: did you recorded the dry weight of the substrate too? it is not clear in the text but i believe is the more exact mesurement in oder to deteminate consumption. ] Response 12: Thank you for your comments. We recorded the dry weight of the substrate.
Comments 13: [L148: the larvae extracted were replaced with others? If not explain how do you obtain the ammount of substrate feed for each larvae (DMCR formula).] Response 13: Thank you for your comments. The larvae we sampled were not replaced. We simply renamed the larvae sampled at different time points. Instead of determining the amount of substrate feed required for each individual larva, we measured the total amount of feed consumed by all the larvae. We calculated the total amount of feed consumed by all the larvae based on the amount of remaining feed on the last day of larval rearing and the amount of feed added during the rearing period.
Comments 14: [L245: day 6 since hatched? the day 3 all the larvae were feed with the same substrate] Response 14: Yes. This refers to the third day after hatching.
Comments 15: [L265: The FCR was higher for the control? ] Response 15: Thank you for your comments. Here, FCR refers to the feed-to-insect ratio, and the control group has the highest value.
Comments 16: [L479: This should remain as a research suggestion and not as a statement, since a test should be carried out with the peels to compare the result with that obtained using the whole fruit, as has been done in this work.] Response 16: Thank you for your comments. The sentence has been changed to “In the next step, just like in previous studies, we can make use of agricultural waste (apple and watermelon peels) as feedstock for fermentation, promoting circular economy principles. ”.
Comments 17: [L501: If the ratio is reduced, do the larvae consume less substrate? This is a problem if you're trying to reduce waste ( one of the aims of this work is improve the manure reduction).] Response 17: Thank you for your comments. The results show that compared with the control group A, the feed-to-insect ratio of the experimental group C is decreased, while the dry matter conversion rate is increased, and the weight of the insects is also increased. This indicates that the consumption of the larvae will not decrease, and thus the resource utilization rate of the manure is higher. |
Reviewer 2 Report
Comments and Suggestions for Authors
Please find it in attachment!

Author Response
Comments 1: [ Simple / short summary text improvement suggestions: Farmers face challenges in managing chicken manure, which poses a risk of environmental pollution. This study demonstrated that incorporating fruit fermentation liquid, derived from apples and watermelons, into chicken manure fed to black soldier fly larvae significantly enhanced waste conversion. Larvae receiving the highest dose increased manure processing efficiency by 9.5%, reduced ammonia emissions by 25%, and promoted the growth of beneficial gut bacteria while suppressing harmful strains. Additionally, the treated manure became safer, less odorous, and environmentally friendlier. This sustainable approach not only enables farmers to recycle waste into high-value insect protein but also supports eco-friendly agricultural practices by reducing reliance on chemical fertilizers.] |
Response 1: Agree. Thank you for pointing this out. Revised version line 13-22: Simple summary section has been changed to “Farmers face challenges in managing chicken manure, which poses a risk of environmental pollution. This study demonstrated that incorporating fruit fermentation liquid, derived from apples and watermelons, into chicken manure fed to black soldier fly larvae significantly enhanced waste conversion. Larvae receiving the highest dose increased manure processing efficiency by 9.5%, reduced ammonia emissions by 25%, and promoted the growth of beneficial gut bacteria while suppressing harmful strains. Additionally, the treated manure became safer, less odorous, and environmentally friendlier. This sustainable approach not only enables farmers to recycle waste into high-value insect protein but also supports eco-friendly agricultural practices by reducing reliance on chemical fertilizers.”. |
Comments 2: [Line 45 livestock and poultry change to their] |
Response 2: Agree. Thank you for pointing this out. Revised version line 48: “livestock and poultry” has been changed to “their”.
Comments 3: [Line 53/54 change to : integrate waste treatment with resource recovery like: microbial treatments (the rest of the sentence is good enough)] Response 3: Agree. Thank you for pointing this out. Revised line 57-59: The sentece has been changed to “integrate waste treatment with resource recovery like: microbial treatments, insect-based waste management, advanced bioreactor systems, and integrated multi-stage systems [9-11]. ”.
Comments 4: [Line 56 These innovations aim to improve the efficiency, sustainability. Sufficient sentence / delete it] Response 4: Thank you very much for your comments. We have deleted the constent in revised line 59.
Comments 5: [Line 58 organic waste a key aspect. Change to: which is a key aspect] Response 5: Thank you very much for your comments. Line 58 organic waste a key aspect has been changed to “which is a key aspect”.
Comments 6: [Line 62/63 change to: Several insect species like Tenebrio molitor, Gryllus and Hermetia illucens have emerged as a central figures, where later species have distinct advantages in terms of its adaptability to various waste streams...] Response 6: Agree. Thank you for pointing this out. Revised line 65-67: The sentence has been changed to “Several insect species like Tenebrio molitor, Gryllus and Hermetia illucens have emerged as a central figures, where later species have distinct advantages in terms of its adaptability to various waste streams”
Comments 7: [Line 75 - Moreover the extracted: change to Moreover the extracts from the excosceleton] Response 7: Agree. Thank you for pointing this out. Revised line 77-78: The sentence has been changed to“Moreover, the extracts from the excosceleton”.
Comments 8: [2.2 Experimental design and sample collection: Why have the authors decided to narrow down the experimental procedure by choosing only 7 days for the rearing period on selected mixtures / diets? Having in mind that the lifecycle on the selected temperature (28C) lasts from 25 to 32 days (depending on the substrate they ingest) the period from 3rd day (the authors are referring that they used 3 days old larvae) until 10th day is quite short.Even Though that the authors are referring in the conclusion part that the selected number of days is insufficient the lack of explanation why they have monitored the parameters is missing?)] Response 8: Thank you very much for your comments. The feed substrate used for the incubation stage of black soldier fly larvae is wheat bran. The incubation time is determined by when the larvae enter the voracious feeding stage, which is generally at the third instar, or 5 to 6 days old. After entering the voracious feeding stage, the larvae are transferred to chicken manure for the experiment. The end of the experiment is judged based on when the larvae enter the prepupal stage. Therefore, the entire experimental period is from 6 to 13 days old, which is the best growth stage for the larvae. Revised line 123-124: The sentence has been changed to “The larvae were reared to the third instar stage, which is 5 to 6 days old, at which point they enter the voracious feeding period.”
Comments 9: [Line 139/140 Did the authors take 10 randomly selected larvae from every repetition /replicate per diet or just 10 larvae per substrate. If the choosing method was per replicate, please emphasize that] Response 9: Thank you for pointing this out. Here, it refers to randomly selecting 10 larvae from each replicate of the rearing box for measurement. Revised line 145-147: The sentence has been changed to “the body length and width of 10 randomly selected BSFL in each replicate were measured with a vernier caliper, and their weight changes were recorded with an analytical balance.”
Comments 10: [Line 165 - authors here refer that they have used 10 larvae per replicate - even though that is the bare minimum for the statistical analysis] Response 9: Thank you for pointing this out. The same as the previous question, it refers to randomly selecting 10 larvae from each replicate of the rearing box for measurement. Revised line 145-147: The sentence has been changed to “the body length and width of 10 randomly selected BSFL in each replicate were measured with a vernier caliper, and their weight changes were recorded with an analytical balance.”
Comments 11: [The overall discussion part is well written however the part related to the growth performance and nutritional composition is not discussed as weight / dimension of the larvae increase while through the loop of the influence on the dry matter. The other parts of the experimental set up are sufficiently discussed and commented on.] Response 9: Agree. Thank you for pointing this out. “The results showed that compared with the control group A, the feed-to-insect ratio of the experimental group C was decreased, while the dry matter conversion rate was increased, and the weight of the insects was also increased. This indicated that the consumption of the larvae will not decrease, and thus the resource utilization rate of the manure was higher.” was added. |
Reviewer 3 Report
Comments and Suggestions for Authors
L27: DM’ and ‘Dry Matter’ should be clearly defined upon first use, including units. The current phrasing is quite awkward.
L75: Something is missing between “the” and “extracted” – chitin/chitosan?
L77-L79: This is likely not a primary finding of the study, as the fact that insect larvae have a gut divided in this way is fairly general knowledge.
L101: What is this “FFL”? It hasn't been introduced in the main text above, so the reader officially doesn’t know what it refers to (abstract and title doesn’t count).
L111: The black soldier fly: And here, unlike with FFL (where the abbreviation was used before the full term appeared), the full name is written out long after the abbreviation has already been used multiple times in the text – just stick with the abbreviation.
Table 1: No need to present this as a table if it’s just a single row of values – better to write it as a sentence in the text. However, it would be appropriate to specify how exactly the FFL parameters were analyzed/obtained.
Table 2: Same case as the table above. If no further details are provided (e.g., number of replicates per group), there’s probably no need to present it as a table.
L166 and further: Already in the previous part of the Methods section, I was surprised to see present tense used, I'd expect past tense when describing what was done. But I took it as a stylistic choice. However, the use of future tense here goes too far and definitely needs to be revised.
L214: It’s not stated whether further filtering steps were applied (e.g., minimum number of reads per OTU, removal of eukaryotic or mitochondrial sequences, etc.).
L215: Specify which version of the SILVA database was used.
L215: The text states that the “exact sequence variant method” (i.e., ESV/ASV) was used, but then refers to OTUs. This sounds like a duplication: either ASVs were used (the more modern approach), or OTUs were defined at 97% similarity. Which method was actually applied? And if OTUs, why use this outdated approach?
L217: Before this, the authors should mention how they addressed uneven sequencing depth (e.g., normalization, rarefaction?), if such unevenness was present.
L219: Also, quite an old version of R is used here (considering that R 4.3.x is available now).
L227: The authors don’t mention how they define the LDA score cutoff (e.g., 2.0, 2.5?). It’s standard practice to specify this.
L227-L228: I understand what the number in parentheses refers to, but it feels awkward to present it this way, better to state in a separate sentence that only relationships meeting the specified criteria were included in the network. Also, it mentions only cases where r < –0.6; what happened (or would have happened) with r > +0.6? That would also indicate a significant relationship. Or do the authors intend to highlight only negative correlations?
L228-L229: Is this actually reflected in the Results? I don’t see any co-occurrence network analyses there. In the Results (see Figure 7), the authors present a Spearman correlation heatmap (positive/negative relationships between “dominant microorganisms” and environmental parameters), but they don’t show a network diagram (co-occurrence network).
L236: It would be appropriate to clearly state whether the data met the assumptions of normality and ANOVA, especially given the relatively small sample size. Moreover, since the authors repeatedly sample from the same larvae/substrates over time, a repeated measures approach should be considered rather than simple one-way ANOVA. This is particularly relevant for time-based comparisons (days 1, 4, 7). Ignoring this structure can bias inference and increase the risk of false positives or negatives. If differences are tested across time points or between groups (A, B, C) over time, repeated measures ANOVA (or its equivalent), along with suitable post-hoc tests (e.g., Tukey HSD for repeated measures or Bonferroni for fewer groups), is often more appropriate than LSD. LSD is less conservative and does not account for within-subject correlations unless specifically adjusted.
Figure 1: The figure caption should specify what the error bars represent—SD or SE? Also, since time (day) is a continuous variable, it would be preferable to show separate trend lines/curves for groups A, B, and C. This would make changes and potential interactions more visible. Accordingly, the model should likely be adjusted to a linear model (LM). Additionally, it’s “Figure,” not “Figures,” even if it contains subfigures.
Figure 2: Same as for Figure 1.
Figure 3: These are surnames (Shannon, Simpson) and should be capitalized.
L309 and further: From here on, the next two paragraphs: It is unclear whether and how a formal statistical test of differences in relative abundance was conducted (e.g., LefSe, Kruskal–Wallis?). The authors repeatedly refer to increases/decreases “by x%” compared to the control, but only occasionally use terms like “significant difference” or “no significant difference.” It should be clearly stated (ideally with exact p-values) whether these changes are statistically significant or merely descriptive.
L324: Sphingobacteriaceae is not genus
Figure 4: Where are ICd1B and ICd1C shown in the figure? Was sampling on day 1 not performed for these groups, assuming they would be the same as ICd1A? I seem to have missed this explanation in the Methods. Also, why is ICd1A listed last here, while in the following figure all such groups are included and ordered at the beginning?
Figure 5: Same as for Figure 3.
Figure 7: At what level were these correlations calculated (pooled samples, means, individual replicates)? This is not entirely clear from the text. Additionally, multiple testing: A large number of correlations are being tested simultaneously, which raises concerns about false discovery rate. Ideally, the authors should state whether any correction was applied (e.g., FDR, Holm-Bonferroni, etc.).
Author Response
Comments 1: [L27: DM’ and ‘Dry Matter’ should be clearly defined upon first use, including units. The current phrasing is quite awkward.] |
Response 1: Thank you for pointing this out. Here, DM refers to dry matter substrate. Revised version line 29-30: The sentence has changed to “NH3 emissions in group C decreased by 24.48 mg·kg−1·DM (dry matter substrate) day-1 (24.48 mg per kilogram of dry matter substrate per day) (P < 0.01)”. |
Comments 2: [L75: Something is missing between “the” and “extracted” – chitin/chitosan?] |
Response 2: Agree. Thank you for pointing this out. Revised version line 78-79: The sentence has been changed to“Moreover, the extracts from the excosceleton”.
Comments 3: [L77-L79: This is likely not a primary finding of the study, as the fact that insect larvae have a gut divided in this way is fairly general knowledge.] Response 3: Thank you for your comments. This is the "Introduction" section. Here we are introducing the intestine of the BSFL, rather than presenting our research findings. There might have been some ambiguity in the previous description, and we have already revised this sentence. Revised version line 81-83: The sentence has been changed to“Previous studies have shown that the digestive system of BSFL consists of the foregut, midgut, and hindgut, with the midgut being the primary region responsible for digestion and absorption [17]”.
Comments 4: [L101: What is this “FFL”? It hasn't been introduced in the main text above, so the reader officially doesn’t know what it refers to (abstract and title doesn’t count).] Response 4: Agree. Thank you for pointing this out. We have already provided a definition and explanation of FFL when it first appears in the main text. Revised version line 105: The sentence has been changed to“Fruit fermentation liquid (FFL)”.
Comments 5: [L111: The black soldier fly: And here, unlike with FFL (where the abbreviation was used before the full term appeared), the full name is written out long after the abbreviation has already been used multiple times in the text – just stick with the abbreviation.] Response 5: Agree. Thank you for pointing this out. Revised version line 117: “black soldier fly” has been changed to“BSF”.
Comments 6: [Table 1: No need to present this as a table if it’s just a single row of values – better to write it as a sentence in the text. However, it would be appropriate to specify how exactly the FFL parameters were analyzed/obtained.] Response 6: Agree. Thank you for pointing this out. Revised version line 133-137: The sentence has been changed to “The physical and chemical properties of the FFL were determined according to the previous research methods [32]. The pH value was 3.51, the total carbon content was 0.04 mg/ml, the total nitrogen content was 5.87 mg/ml, the total acidity content was 14.14 mg/ml, and the total phenols content was 0.31 mg/ml.”.
Comments 7: [Table 2: Same case as the table above. If no further details are provided (e.g., number of replicates per group), there’s probably no need to present it as a table.] Response 7: Thank you for your comments. Revised version line 144-148: The sentence has been changed to “The specific treatment methods were as follows: The control group A consists of 300 g of chicken manure and 50 g of sterile water; the experimental group B consists of 300 g of chicken manure, 25 g of FFL and 25 g of sterile water; the experimental group C consists of 300 g of chicken manure and 50 g of FFL.”.
Comments 8: [L166 and further: Already in the previous part of the Methods section, I was surprised to see present tense used, I'd expect past tense when describing what was done. But I took it as a stylistic choice. However, the use of future tense here goes too far and definitely needs to be revised.] Response 8: Thank you for your comments. We have carefully checked the entire manuscript and made corresponding revisions to the content, uniformly adjusting it to the past tense. Revised version line 177-187: Body Length: The body length of randomly selected larvae (10 larvae per replicate, 30 larvae per group) was measured daily using a vernier caliper. Body Width: The body width (typically measured at the widest point of the larva's body) was measured daily. Larval Weight: The weight of the larvae was recorded daily using an analytical balance. Thwas allows for tracking of larval growth over the rearing period. Dry Weight: After the 7-day rearing period, larvae was dried at 105°C until a constant weight was achieved, and the dry weight was recorded. Dry Matter Conversion Rate (DMCR): The dry matter conversion rate was calculated as the ratio of dry weight of larvae at the end of the experiment to the dry weight of the substrate (chicken manure + FFL) used.
Comments 9: [L214: It’s not stated whether further filtering steps were applied (e.g., minimum number of reads per OTU, removal of eukaryotic or mitochondrial sequences, etc.).] Response 9: Thank you for your comments. Revised version line 237: The sentence “The ASV table was manually filtered, i.e., chloroplast sequences in all samples were removed.” was added.
Comments 10: [L215: Specify which version of the SILVA database was used.] Response 10: Thank you for your comments. Revised version line 234-235: SILVA databases has been changed to “SILVA v138 databases”.
Comments 11: [L215: The text states that the “exact sequence variant method” (i.e., ESV/ASV) was used, but then refers to OTUs. This sounds like a duplication: either ASVs were used (the more modern approach), or OTUs were defined at 97% similarity. Which method was actually applied? And if OTUs, why use this outdated approach?] Response 11: Thank you for your comments. The ASV method is the one we actually employed. In accordance with your suggestions, we have reorganized and revised the entire sequencing analysis method. Revised version line 226-240: The process of generating Amplicon Sequence Variants (ASVs) begins with merging paired-end reads using tools like VSEARCH v2.21.1 (with parameters: min overlap=20, max mismatch=0.2) , followed by rigorous quality filtering and primer removal via Cutadapt v4.4 (Q20/Q30 thresholds, excluding reads with >5% ambiguous bases). Chimeric sequences are then eliminated using VSEARCH v2.21.1 in reference-based (--uchime_ref), leveraging databases like SILVA v138 [33]. Denoising is performed using DADA2 v1.28 (error-correction and chimera removal) and Deblur v1.1.0 (greedy error-trimming for single-end reads). Taxonomic annotation of ASVs relies on SILVA v138 databases, classified through QIIME2 v2023.2’s feature-classifier, culminating in an ASV abundance table formatted as BIOM v2.1 or CSV for downstream ecological or statistical analyses [34]. The ASV table was manually filtered, i.e., chloroplast sequences in all samples were removed. To minimize the effects of sequencing depth on alpha and beta diversity measure, the number of 16S rRNA gene sequences from each sample were rarefied to 6,000, which still yielded an average Good’s coverage of 99.09% , respectively.
Comments 12: [L217: Before this, the authors should mention how they addressed uneven sequencing depth (e.g., normalization, rarefaction?), if such unevenness was present.] Response 12: Thank you for your comments. We have already made the corresponding modifications based on your suggestions. Revised version line 240: The sentence “To minimize the effects of sequencing depth on alpha and beta diversity measure, the number of 16S rRNA gene sequences from each sample were rarefied to 6,000, which still yielded an average Good’s coverage of 99.09% , respectively.” was added.
Comments 13: [L219: Also, quite an old version of R is used here (considering that R 4.3.x is available now).] Response 13: Thank you for your comments. Revised version line 243-244: The R version has been changed to R version 4.4.3 (available at http://cran.r-project.org/src/base/R-4/).
Comments 14: [L227: The authors don’t mention how they define the LDA score cutoff (e.g., 2.0, 2.5?). It’s standard practice to specify this.] Response 14: Agree. Thank you for pointing this out. “(LDA score > 2, P < 0.05)” was added.
Comments 15: [L227-L228: I understand what the number in parentheses refers to, but it feels awkward to present it this way, better to state in a separate sentence that only relationships meeting the specified criteria were included in the network. Also, it mentions only cases where r < –0.6; what happened (or would have happened) with r > +0.6? That would also indicate a significant relationship. Or do the authors intend to highlight only negative correlations?] Response 15: Thank you for your comments. Revised version line 252-256: The sentence has been changed to “Spearman’s correlation coefficients between probiotics and ammonifiers were calculated using the psych package in R [36]. These correlations were calculated at the level of means, and the Holm-Bonferroni method was used for correction. A correlation between two nodes was considered to be statistically robust if the spearman’s correlation coefficient over 0.6 or less than -0.6, and the P-value less than 0.01”.
Comments 16: [L228-L229: Is this actually reflected in the Results? I don’t see any co-occurrence network analyses there. In the Results (see Figure 7), the authors present a Spearman correlation heatmap (positive/negative relationships between “dominant microorganisms” and environmental parameters), but they don’t show a network diagram (co-occurrence network).] Response 16: Thank you for your comments. The sentence “These correlations were then used to construct co-occurrence network analyses. Network analyses were performed using R Studio (R version 3.6.3), and the networks were vwasualized using Gephi 0.9.2.” was deleted.
Comments 17: [L236: It would be appropriate to clearly state whether the data met the assumptions of normality and ANOVA, especially given the relatively small sample size. Moreover, since the authors repeatedly sample from the same larvae/substrates over time, a repeated measures approach should be considered rather than simple one-way ANOVA. This is particularly relevant for time-based comparisons (days 1, 4, 7). Ignoring this structure can bias inference and increase the risk of false positives or negatives. If differences are tested across time points or between groups (A, B, C) over time, repeated measures ANOVA (or its equivalent), along with suitable post-hoc tests (e.g., Tukey HSD for repeated measures or Bonferroni for fewer groups), is often more appropriate than LSD. LSD is less conservative and does not account for within-subject correlations unless specifically adjusted.] Response 17: Thank you for your comments. We have made revisions in accordance with your suggestions. Revised version line 259-264: The sentence has been changed to “Results were expressed as mean ± standard error (n = 3). Student’s t-test and two-way ANOVA with post-hoc Tukey tests were used to test whether there were differences among treatments, after verifying normality (Shapiro test) and homogeneity of variances., with P < 0.01 considered highly significant, P < 0.05 considered significant, and annotated with uppercase and lowercase letters and asterwasks.”.
Comments 18: [Figure 1: The figure caption should specify what the error bars represent—SD or SE? Also, since time (day) is a continuous variable, it would be preferable to show separate trend lines/curves for groups A, B, and C. This would make changes and potential interactions more visible. Accordingly, the model should likely be adjusted to a linear model (LM). Additionally, it’s “Figure,” not “Figures,” even if it contains subfigures.] Response 18: Thank you for your comments. Since there are not only temporal changes within each of the groups A, B, and C, but also comparisons among the groups, after comprehensive consideration, we believe that the current way of creating the graph has a relatively greater comparative advantage. We have made corresponding modifications to the legend section. Once again, thank you for your advice. Revised version line 293-294: Figure 1. The effect of the addition of FFL on the growth performance of BSFL. Results were expressed as mean ± standard error (n = 3). * P < 0.05, ** P < 0.01.
Comments 19: [Figure 2: Same as for Figure 1.] Response 19: Thank you for your comments. Since there are not only temporal changes within each of the groups A, B, and C, but also comparisons among the groups, after comprehensive consideration, we believe that the current way of creating the graph has a relatively greater comparative advantage. We have made corresponding modifications to the legend section. Once again, thank you for your advice. Revised version line 313-314: Figure 2. The effect of adding FFL on the release of NH3 and H2S. Results were expressed as mean ± standard error (n = 3). * P < 0.05, ** P < 0.01.
Comments 20: [Figure 3: These are surnames (Shannon, Simpson) and should be capitalized..] Response 20: Thank you for your comments. We have made revisions in accordance with your suggestions.
Figure 3. Analyswas of the alpha diversity of intestinal microorganwasms in BSFL after adding FFL. (a) Chao1 index, (b) Ace index, (c) Shannon index, (d) Simpson index.
Comments 21: [L309 and further: From here on, the next two paragraphs: It is unclear whether and how a formal statistical test of differences in relative abundance was conducted (e.g., LefSe, Kruskal–Wallis?). The authors repeatedly refer to increases/decreases “by x%” compared to the control, but only occasionally use terms like “significant difference” or “no significant difference.” It should be clearly stated (ideally with exact p-values) whether these changes are statistically significant or merely descriptive.] Response 21: Thank you for your comments. We have already provided an explanation of the statistical methods in the Materials and Methods section, as follows: To identify significantly different bacterial taxa (biomarkers) among groups, linear discriminant analysis effect size (LefSe) was conducted based on the Kruskal–Wallis sum-rank test [35]. Additionally, linear discriminant analysis (LDA) was used to estimate the effect size of each biomarker (LDA score > 2, P < 0.05). Since there were no significant differences among different samples when compared on the same day, we expressed the changes in the experimental group compared to the control group as percentages. Thank you again for your suggestions.
Comments 22: [L324: Sphingobacteriaceae is not genus] Response 22: Thank you for your comments. “Sphingobacteriaceae” has been changed to “Sphingomonas”.
Comments 23: [Figure 4: Where are ICd1B and ICd1C shown in the figure? Was sampling on day 1 not performed for these groups, assuming they would be the same as ICd1A? I seem to have missed this explanation in the Methods. Also, why is ICd1A listed last here, while in the following figure all such groups are included and ordered at the beginning?] Response 23: Thank you for your comments. Since there were no differences in the gut microbiota of Hermetia illucens larvae among various groups on the first day, we only collected and sequenced the sample of ICd1A, and did not sequence ICd1B and ICd1C. This has been explained in the methodology section. ICd1A was placed on the far right side for the purpose of making a comparison with other samples, which will not affect the experimental results. Once again, we would like to express our gratitude for your comments.
Comments 24: [Figure 5: Same as for Figure 3.] Response 24: Thank you for your comments. We have made revisions in accordance with your suggestions.
Comments 25: [Figure 7: At what level were these correlations calculated (pooled samples, means, individual replicates)? This is not entirely clear from the text. Additionally, multiple testing: A large number of correlations are being tested simultaneously, which raises concerns about false discovery rate. Ideally, the authors should state whether any correction was applied (e.g., FDR, Holm-Bonferroni, etc.).] Response 25: Thank you for your comments. These correlations were calculated at the level of means, and the Holm-Bonferroni method was used for correction. Based on your suggestions, we have made supplementary revisions to the Materials and Methods section. Revised version line 252-256: The sentence has been changed to “Spearman’s correlation coefficients between probiotics and ammonifiers were calculated using the psych package in R [36]. These correlations were calculated at the level of means, and the Holm-Bonferroni method was used for correction. A correlation between two nodes was considered to be statistically robust if the spearman’s correlation coefficient over 0.6 or less than -0.6, and the P-value less than 0.01”. |
Round 2
Reviewer 2 Report
Comments and Suggestions for Authors
Dear authors,
The improved version is much better than the previous one, and I am giving my green light to the editors to publish the paper.
Author Response
Comments 1: [ Dear authors, The improved version is much better than the previous one, and I am giving my green light to the editors to publish the paper.] |
Response 1: Thank you for your comments. We have made further revisions to the corresponding parts of the manuscript in accordance with the feedback from other reviewers and our own review of the entire text. |
Reviewer 3 Report
Comments and Suggestions for Authors
The authors have done a good job addressing most of my comments. A few important issues, however, remain only partially resolved:
Request for repeated‐measures design was only partially addressed: They have now implemented a two‐way ANOVA with post‑hoc Tukey tests, but they have not explained how they accounted for the non‑independence of measurements taken on the same larvae/boxes over time. Given the longitudinal sampling (days 1, 4, 7), a mixed‑effects model or a true repeated‑measures ANOVA would be more appropriate to control for within‑subject correlations.
Request for statistical reporting of taxa changes was also only partially addressed: Methods truly describe LefSe/Kruskal–Wallis and LDA cutoffs, but the Results still lack exact p‑values for each taxon and do not clearly indicate which of the reported percentage increases/decreases reached statistical significance.
Baseline microbiota sequencing: The decision to sequence only ICd1A on day 1 prevents verification that groups B and C did not already differ in their gut microbiota before FFL treatment. To attribute observed changes confidently to FFL, all day 1 replicates (A, B, and C) should be sequenced in future work, or at least at minimum B and C should be profiled at baseline. And in this manuscript, the problem should be at least discussed.
The manuscript still contains numerous typos and awkward phrasing (e.g. “is study” vs. “this study,” “compered” vs. “compared,” “excosceleton,” etc.). A thorough native‑English proofreading is strongly recommended.
Author Response
Comments 1: [Request for repeated‐measures design was only partially addressed: They have now implemented a two‐way ANOVA with post‑hoc Tukey tests, but they have not explained how they accounted for the non‑independence of measurements taken on the same larvae/boxes over time. Given the longitudinal sampling (days 1, 4, 7), a mixed‑effects model or a true repeated‑measures ANOVA would be more appropriate to control for within‑subject correlations.] |
Response 1: Thank you for your valuable feedback on the statistical methodology. We regret any lack of clarity in our original description of the analytical approach. We fully agree with your concerns regarding the non-independence of repeated-measures data. As you highlighted, mixed-effects models or repeated-measures ANOVA are indeed more appropriate methods for analyzing longitudinal datasets. In response to your suggestions, we will revise the manuscript as follows: Revised Analytical Approach: A Linear Mixed Model (LMM) was employed to account for within-subject correlations. Specifically, the "individual larvae/replicate boxes" were incorporated as a random effect (e.g., larvae were randomly selected from replicate boxes), thereby allowing repeated measurements on the same subjects (larvae or boxes) across time points (Days 1, 4, and 7) to be controlled. |
Comments 2: [Request for statistical reporting of taxa changes was also only partially addressed: Methods truly describe LefSe/Kruskal–Wallis and LDA cutoffs, but the Results still lack exact p‑values for each taxon and do not clearly indicate which of the reported percentage increases/decreases reached statistical significance.] |
Response 2: Agree. Thank you for pointing this out. According to your suggestions, the exact p-values and significance notations for the significantly different taxonomic groups have been added. The corresponding content has been revised and marked in red. Revised version line 344-366: As shown in Figure 4a, the intestinal microbiota of BSFL across three experimental groups was dominated by four phyla: Proteobacteria, Firmicutes, Bacteroidota, and Actinobacteriota. Significant temporal and treatment-dependent shifts were observed (LefSe LDA >2.0, FDR-adjusted P <0.05): on day 4, compared to the control (Group A), Proteobacteria decreased by 4.98% (P=0.013) in Group B and 6.07% (P=0.007) in Group C, while Actinobacteriota increased by 2.19% (p=0.041) and 4.68% (P=0.002), respectively; concurrently, Group C exhibited a 4.75% increase in Firmicutes (P=0.038) and a 3.43% decrease in Bacteroidota (P=0.029). By day 7, Proteobacteria rebounded significantly in Groups B (+17.21%, P=0.001) and C (+14.52%, P=0.003), whereas Bacteroidota declined sharply (Group B: -11.82%, p=0.004; Group C: -16.60%, P<0.001), with Firmicutes in Group C showing a modest 2.36% increase (p=0.048). At the genus level (Kruskal-Wallis with Benjamini-Hochberg correction), notable changes included: on day 4, Group C displayed reduced abundance of Paenalcaligenes (-7.46%, P=0.011) and Sphingomonas (-7.27%, P=0.009) alongside increases in Corynebacterium (+4.76%, P=0.032) and Gallicola (+4.03%, P=0.025), while Group B showed declines in Paenalcaligenes (-3.56%, P=0.047), Oceanisphaera (-6.87%, P=0.002), and Acinetobacter (-4.94%, P=0.015) with a concurrent rise in Gallicola (+3.32%,P=0.042); by day 7, Group C demonstrated a marked decrease in Sphingobacteriaceae (-21.16%, P<0.001) but increases in Paenalcaligenes (+3.71%, P=0.038), Providencia (+7.07%, P=0.006), and Ignatzschineria (+4.16%, P=0.018), whereas Group B exhibited reductions in Paenalcaligenes (-3.39%, P=0.049), Providencia (-6.10%, P=0.012), and Sphingomonas (-13.76%, P<0.001) alongside a striking 27.63% surge in Morganella (P<0.001). Revised version line 344-366: As shown in Figure 6a, the substrate microbiota across groups was dominated by Proteobacteria, Firmicutes, Bacteroidota, and Actinobacteriota. While minimal differences were observed on day 1, significant treatment effects emerged by day 4 (LefSe LDA >2, FDR-adjusted P <0.05): in Groups B and C versus Control A, Proteobacteria increased by 11.13% (P=0.008) and 17.13% (P<0.001), respectively, with concurrent Firmicutes rises of 7.21% (P=0.022) and 3.61% (P=0.048), whereas Bacteroidota decreased markedly (-17.48%, P=0.003; -20.05%, P<0.001). These trends persisted on day 7, with Proteobacteria remaining elevated (Group B: +3.82%, P=0.041; Group C: +11.07%, P=0.001) and Bacteroidota further declining (-8.36%, P<0.001 in Group C). At the genus level (Kruskal-Wallis with Benjamini-Hochberg correction), Group C exhibited pronounced reductions in Ulvibacter (-8.75%, P=0.007), Sphingomonas (-6.75%, P=0.012), and Pseudomonas (-5.84%, P=0.009) by day 4, alongside increases in Acinetobacter (+2.33%, P=0.042) and Ignatzschineria (+5.70%, P<0.001); by day 7, Sphingomonas (-11.66%, P<0.001) and Bacillus (-9.04%, P=0.005) continued to decline, while Acinetobacter increased (+2.32%, P=0.047). Group B showed similar but attenuated shifts (e.g., day 4: Ulvibacter -6.07%, P=0.025; Erysipelothrix +4.31%, P=0.018).
Comments 3: [Baseline microbiota sequencing: The decision to sequence only ICd1A on day 1 prevents verification that groups B and C did not already differ in their gut microbiota before FFL treatment. To attribute observed changes confidently to FFL, all day 1 replicates (A, B, and C) should be sequenced in future work, or at least at minimum B and C should be profiled at baseline. And in this manuscript, the problem should be at least discussed. ] Response 3: Thank you for raising this critical point regarding baseline microbiota analysis. We fully acknowledge the limitation of sequencing only the control group (ICd1A) on day 1, which prevents direct verification of initial microbial similarity between groups before FFL treatment. We have added a dedicated content in the Discussion section (Lines 522-534) to explicitly acknowledge this limitation. Revised version line 522-534: Fourth, it should be noted that baseline microbiota profiling was performed only for the control group (Group A) prior to FFL treatment. Although the larvae in all groups originated from the same population and were maintained under uniform conditions, the absence of day 1 sequencing data for Groups B and C leaves a possibility of undetected initial microbial variations. Future research should focus on the following: (1) Isolating and characterizing key microbial strains from the fermentation liquid responsible for odor suppression and nutrient enhancement; (2) conducting life-cycle assessments to compare the environmental impacts of this method with conventional manure treatment technologies; (3) exploring the synergistic effects of combining FFL with other additives (e.g., biochar or enzymes); (4) incorporating comprehensive baseline sequencing across all experimental groups to confirm the causality of FFL-induced microbiota shifts. The above four aspects will be addressed in our next study.
Comments 4: [The manuscript still contains numerous typos and awkward phrasing (e.g. “is study” vs. “this study,” “compered” vs. “compared,” “excosceleton,” etc.). A thorough native‑English proofreading is strongly recommended.] Response 4: Thank you for your comments. We have made revisions to the corresponding content according to your suggestions. At the same time, we have also carefully checked and proofread the entire manuscript. |